# Analysis of intestinal epithelial cell responses to *Cryptosporidium* highlights the temporal effects of IFN-γ on parasite restriction

**Ryan D. Pardy**, **Katelyn A. Walzer, Bethan A. Wallbank, Jessica H. Byerly, Keenan M. O'Dea, Ian S. Cohn, Breanne E. Haskins, Justin L. Roncaioli, Eleanor J. Smith, Gracyn Y. Buenconsejo, Boris Striepen, Christopher A. Hunter** *

Department of Pathobiology, School of Veterinary Medicine, University of Pennsylvania, Philadelphia, Pennsylvania, United States of America

* chunter@vet.upenn.edu

**Data Availability Statement:** All data generated or analyzed during this study are included in this published article and its supporting information

## Abstract

The production of IFN-γ is crucial for control of multiple enteric infections, but its impact on intestinal epithelial cells (IEC) is not well understood. *Cryptosporidium* parasites exclusively infect epithelial cells and the ability of interferons to activate the transcription factor STAT1 in IEC is required for parasite clearance. Here, the use of single cell RNA sequencing to profile IEC during infection revealed an increased proportion of mid-villus enterocytes during infection and induction of IFN-γ-dependent gene signatures that was comparable between uninfected and infected cells. These analyses were complemented by *in vivo* studies, which demonstrated that IEC expression of the IFN-γ receptor was required for parasite control. Unexpectedly, treatment of *Ifng*[-/-] mice with IFN-γ showed the IEC response to this cytokine correlates with a delayed reduction in parasite burden but did not affect parasite development. These data sets provide insight into the impact of IFN-γ on IEC and suggest a model in which IFN-γ signalling to uninfected enterocytes is important for control of *Cryptosporidium*.

## Author summary

The cytokine interferon-gamma (IFN-γ) plays an important role in the control of intracellular infections by a wide variety of bacteria, viruses and parasites. While the impact of IFN-γ on immune cells has been a major research focus, how it impacts intestinal epithelial cells (IEC) remains poorly understood. *Cryptosporidium* parasites are an important cause of morbidity in a variety of epidemiological settings and exclusively infect IEC. Recent advances in the ability to genetically modify and study *Cryptosporidium* in wild-type hosts provides a useful model to investigate IEC-intrinsic mechanisms of pathogen control. In this study, single cell RNA-sequencing was used to analyze the IEC response to infection and IFN-γ signalling. This analysis demonstrates that during infection there are broad changes in the IEC compartment that include the robust induction of IFN-γ-stimulated genes. In addition, infected IEC remain responsive to IFN-γ signalling, and this

files. RNA sequencing data has been deposited to the GEO repository (GSE246500).

**Funding:** This work was funded by the National Institutes of Health (R01-AI148249; https://www.nih.gov/). R.D.P. is supported by a Fellowship award from the Canadian Institutes of Health Research (MFE-176621; https://cihr-irsc.gc.ca/e/193.html) and a Postdoctoral Training award from the Fonds de Recherche du Québec – Santé (300355; https://frq.gouv.qc.ca/en/). B.E.H is supported by training grant T32AI007532, K.M.O. is supported by training grant T32AI055428 and I.S.C. is supported by fellowship F30AI169744. K.A.W. is supported by a postdoctoral fellowship from the National Institutes of Health (F32 AI154666). The funders had no role in study design, data collection and analysis, decision to publish, or preparation of the manuscript.

**Competing interests:** The authors have no competing interests to declare.

cytokine causes a delayed reduction in parasite burden that correlates with the kinetics of IEC responsiveness to IFN-γ. Together, help define the relationship between the kinetics of IFN-γ responsiveness and control of *Cryptosporidium* in IEC.

## Introduction

Intestinal epithelial cells (IEC) are infected by a wide range of viruses, bacteria, and parasites. While many of these micro-organisms use IEC as an initial site of replication, or a "portal" through which they establish systemic infection, a subset of these pathogens is restricted to this cellular niche and remains in the small intestine. The most common cells in the intestinal epithelium are enterocytes, tall columnar cells that mediate nutrient uptake and express antimicrobial peptides, as well as specialized IEC subsets that include goblet cells and Paneth cells, which can function to limit pathogen invasion [1]. The ability of the immune system to promote IEC-intrinsic anti-microbial mechanisms provides an additional important contribution to pathogen clearance. Thus, the cytokine IFN-γ has a crucial role to stimulate host cells to restrict infection by a variety of IEC-tropic pathogens, which includes bacteria (*Listeria monocytogenes*, *Salmonella*), viruses (rotavirus), and the parasites *Cryptosporidium* spp. and *Toxoplasma gondii* [2–5]. Although IFN-γ is known to impact various facets of IEC biology [6], how IFN-γ mediates pathogen control in IEC is not well understood.

*Cryptosporidium* species are apicomplexan parasites that are an important cause of diarrheal disease in several epidemiological settings [7]. Most human cases of cryptosporidiosis are caused by *Cryptosporidium hominis* or *Cryptosporidium parvum* and while symptomatic infections are typically self-limiting, in immunocompromised populations this diarrheal disease can be life threatening [8]. Infection is initiated by ingestion of oocysts, which release sporozoites in the small intestine that then invade IEC, where they reside in a parasitophorous vacuole (PV) at the apical tip of the cell [9]. The parasites undergo three asexual cycles of replication and infection before differentiation into male and female parasites [10,11]. The release of males allows fertilization of intracellular females and production of new oocysts that are shed in the feces, or that can excyst within the host and thereby maintain infection. Each cycle of asexual replication and reinvasion is approximately 12 hours and is associated with death of infected cells and *in vivo* alterations to tight and adherens junctions that contribute to the severe watery diarrhea observed in humans and livestock [5]. Initial studies using *C. parvum* identified the importance of T cells and the cytokines IFN-γ, IL-12, and IL-18 for infection control [3,12–15]. Subsequently, it has been shown that IL-12 has a crucial role in the polarization of T helper 1 cells [16,17], while local IL-12 and IL-18 drive type 1 innate lymphoid cell (ILC1) and natural killer cells to produce IFN-γ [18]. The dominant role of IFN-γ in the response to *Cryptosporidium* is evident from infections in mice deficient for this cytokine, which result in high parasite burden and a failure to clear the infection [3,19]. Multiple mechanisms of IFN-γ-dependent control have been described for other intracellular pathogens [2] but it is unclear whether these pathways are induced in IEC or are capable of restriction of *Cryptosporidium*, or whether the parasite can antagonize the effects of IFN-γ.

Several studies have demonstrated that *Cryptosporidium* infection leads to the production of types I, II, and III IFN *in vivo*, each of which can act on IEC to signal through the transcription factor signal transducer and activator of transcription 1 (STAT1) [18,20–22]. While the early production of IFN-λ provides a transient mechanism of resistance to *Cryptosporidium* [20,21], the contribution of type I IFN is less clear with a literature that suggests it is protective [23] or that it supports parasite replication *in vivo* [20,22]. In contrast, there is a consensus that

IFN-γ is a major mediator of resistance to *Cryptosporidium* required for parasite clearance [3,12,19,24]. The ability of the immune system to promote control of *Cryptosporidium* is dependent on STAT1 expression in IEC and, to a lesser extent, IFN-γ-dependent induction of immunity-related GTPase m1 and m3 (Irgm1 and Irgm3) [18]. Since mice with an IEC-specific deletion of STAT1 (*Stat1*$^{\Delta IEC}$ mice) have a similar susceptibility to *C. parvum* infection as *Ifng*$^{-/-}$ mice [18], these observations suggest that IFN-γ has a dominant role in the immune response to *Cryptosporidium*. However, it has been a challenge to define the contribution of IFN-γ (versus other IFNs) on IEC *in vivo* to promote parasite control [5].

In this study, the impact of IFN-γ signalling on IEC, and its role in restriction of *C. parvum* infection, were investigated. Single cell RNA-sequencing (scRNA-seq) revealed global alterations to the IEC compartment during infection, which included widespread induction of IFN-γ-stimulated gene expression. Additional scRNA-seq and flow cytometry-based experiments demonstrated that infected IEC remain responsive to IFN-γ and that this response peaks 12-24h post-exposure. These kinetics correlated with *in vivo* observations that the administration of IFN-γ led to a robust, but delayed, decrease in oocyst shedding. Together, these studies provide insight into the global impact of infection with *Cryptosporidium* on IEC and suggest a model in which IFN-γ signalling to uninfected enterocytes is important to limit the *Cryptosporidium* growth cycle within its host.

## Results

### Single cell transcriptomic analysis of IEC responses during *Cryptosporidium* infection

To gain a global understanding of how *Cryptosporidium* infection impacts IEC *in vivo*, mice were infected with a mouse-adapted strain of *C. parvum* (maCp) that expresses mCherry and nanoluciferase (nLuc) [18] and scRNA-seq was performed on cells from the ileum. For these experiments, the epithelial fraction, which includes IEC and intraepithelial lymphocytes (IEL), was extracted from uninfected mice and those at 4- or 10-days post-infection (dpi); 4 dpi is a timepoint with detectable oocyst shedding and IFN-γ production while by day 10 infection is nearly resolved [18]. Across samples a total of 24,302 cells were sequenced and after quality control there remained 5,411 cells from the uninfected sample, 7,309 from 4 dpi, and 5,290 from 10 dpi. Following data normalization, cells were partitioned into 24 clusters. Uniform Manifold Approximation and Projection (UMAP) visualization of these clusters and marker gene expression (S1 Table and S1A–S1C Fig) were used to identify prominent *Epcam*$^+$ IEC (clusters 1–13) and *Ptprc*$^+$ (CD45$^+$) IEL (clusters 14–19), which are shown in Fig 1A.

Analysis of IEC for cluster-specific marker gene expression (S2 Table) revealed that the majority of cells sequenced were enterocytes that separated into 8 sub-clusters (clusters 2–9), but also included intestinal stem cells/transit amplifying cells (cluster 1), goblet cells (cluster 10), tuft cells (cluster 11), enteroendocrine cells (cluster 12) and Paneth cells (cluster 13) (Figs 1A and S1D–S1H). Monocle3 trajectory analysis was used to order enterocyte and goblet cell clusters in pseudotime, with *Olfm4*$^+$ intestinal stem cells set as the origin (Figs 1B and 1C and S1G). This approach showed that pseudotime values increased from clusters 1–9, implying that cells with low pseudotime values are nearest the crypt and villus base, while those with the highest values resided at the villus tip (Fig 1B and 1C). Comparison of the expression of marker genes for crypt-adjacent, early mid-villus, late mid-villus and villus tip enterocytes from a recent study of enterocyte zonation [25] against pseudotime was used to verify that this analysis reflected enterocyte progression along the villus (S1I–S1M Fig). Thus, this dataset provides coverage of the developmental stages of IEC and allows the placement of different clusters along the villus axis, as represented in Fig 1C.

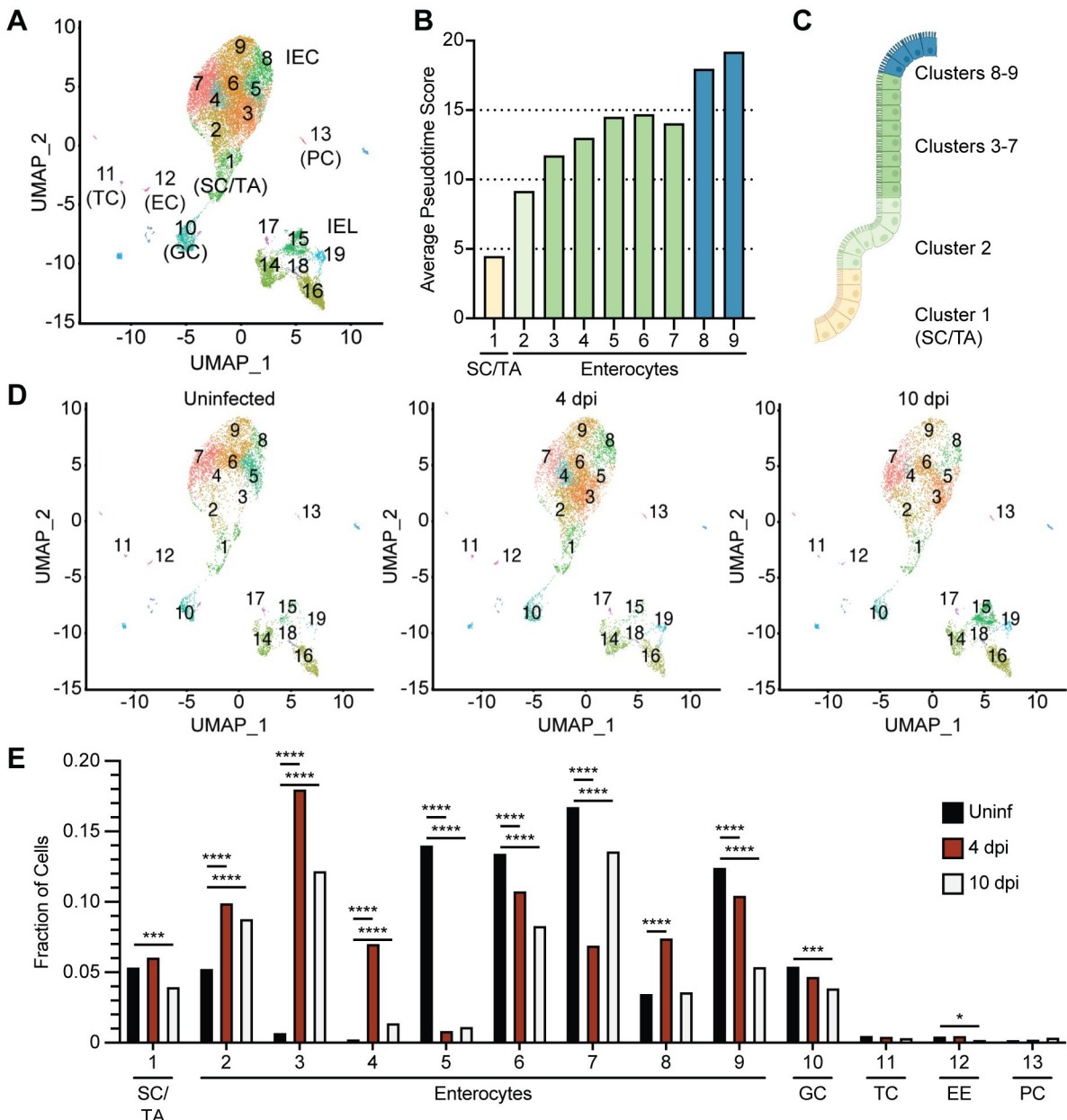

**Fig 1. scRNA-seq of IEC during *Cryptosporidium* infection.** (**A**) Uniform Manifold Approximation and Projection (UMAP) of scRNA-seq of IEC and IEL from uninfected WT mice and WT mice 4 and 10 dpi with maCp. (**B**) Average pseudotime score for each enterocyte cluster. (**C**) Cartoon depicting where cells from each enterocyte cluster are projected to lie in the crypt and villus, coloured by pseudotime value. (**D**) UMAP separated by sample. (**E**) Fraction of cells in IEC clusters separated by sample. Abbreviations: stem cells/transit amplifying cells (SC/TA); goblet cells (GC); tuft cells (TC); enteroendocrine cells (EC); Paneth cells (PC); IFN-γ stimulated gene (ISG). Data in (**E**) were analyzed by $\chi^2$ test between uninfected and 4 dpi and between uninfected and 10 dpi. *$p < 0.05$, ***$p < 0.001$, ****$p < 0.0001$.

To determine how cell states changed over the course of infection, individual UMAP projections were plotted for each infection time point (Fig 1D). Relative to uninfected mice, IEL clusters 14–18 were comparable or decreased 4 dpi, while IEL cluster 19 showed a steady increase over time and clusters 15 and 16 were expanded at 10 dpi (Figs 1D and S1N). These increases reflected recruitment of canonical CD4[+] and CD8αβ[+] T cells to the epithelium and expansion of resident CD8αα[+] T cells (S1O Fig and S1 Table). In contrast, the proportion of

stem cells, tuft cells, and Paneth cells were comparable across timepoints, while there were small decreases to the goblet cell and enteroendocrine cell populations 10 dpi (Fig 1D and 1E). In uninfected mice, the enterocyte populations were composed of clusters 2, 5, 6, 7, 8 and 9 but by 4 dpi the IEC population was characterized by the loss of cluster 5, the emergence of clusters 3 and 4, and a transient expansion of cluster 8 (Fig 1D and 1E). At 10 dpi, most enterocyte clusters were comparable in frequency to uninfected mice, although cluster 3 remained expanded (Fig 1D and 1E). Additional clusters were either distributed comparably across each time point or contained few cells, which precluded in-depth analysis (Figs 1D and S1N). Thus, *Cryptosporidium* infection leads to transient changes in the enterocyte populations in the middle and tip of the villi that resolve as infection is controlled.

## IFN-γ signals to IEC to promote control of *Cryptosporidium* infection

As an unbiased measure to investigate the IEC pathways that are altered during infection, differential gene expression analysis was applied to identify the top 100 marker genes that distinguish the enterocyte and goblet cell clusters (S2 Table). While there is some overlap of gene expression between enterocytes, this approach highlighted distinct transcriptional profiles of each cluster (S2A Fig and S2 Table). Use of the Database for Annotation, Visualization and Integrated Discovery (DAVID) identified significantly enriched functional annotations that were separated into five categories: Cell Structure, Cellular Processes, Immune Response, Mitochondria, and Nutrient/Ion Transport (S2 Table). Genes annotated with functions in nutrient and ion transport were expressed across clusters 3–5 and 7–9, which suggests these pathways are not differentially regulated during infection (S2E Fig). Intestinal stem cells and transit amplifying cells (cluster 1) showed enhanced expression of genes related to mitochondria and energy production (S2C Fig), which has been previously associated with homeostatic cell turnover within this niche [26]. The goblet cell cluster stood out as being enriched for genes related to the unfolded protein response (S2B Fig), a cellular response associated with mucus secretion [27]. Immune response pathways related to antigen processing and presentation and IFN-γ responsiveness were upregulated in enterocyte clusters 3, 4, and 8, which are expanded 4 dpi (hereafter referred to as IFN-γ-stimulated gene [ISG] enterocytes) (Fig 2A). Additional functional annotations enriched in these ISG enterocytes included genes related to the actin cytoskeleton and cellular organization (S2D Fig and S2 Table), processes in which IFN-γ may also play a role [28].

Previous bulk RNA sequencing analysis identified seven ISGs (*Gbp7*, *Ifi47*, *Igtp*, *Iigp1*, *Irgm1*, *Tgtp1*, and *Zbp1*) that were upregulated in IEC during maCp infection [18]. These genes were combined to generate an IFN-γ response score, which is calculated as the expression of the seven ISGs minus expression of a control gene set (randomly selected genes that are expressed comparably between clusters). Positive IFN-γ response scores reflect upregulation of ISGs, scores near 0 indicate expression that is not significantly different than the control gene set, and negative values reflect little to no expression in the cluster. In uninfected mice, these genes are largely silent while at 4 dpi infection there was significantly increased IFN-γ-response score in goblet cells and ISG enterocytes (i.e. clusters 3, 4 and 8) which had largely resolved 10 dpi (Figs 2B and S2F). A recent study identified that a set of canonical IFN-γ-stimulated genes were upregulated in mice 2 dpi with *C. parvum* [22] and these were also enhanced in our data sets (S2G Fig). Furthermore, when single-sample gene set enrichment analysis (ssGSEA) was conducted for the HALLMARK_INTERFERON_GAMMA_RESPONSE gene set it affirmed the presence of an IFN-γ-dependent gene signature in the infection-associated ISG enterocytes and goblet cells (Fig 2C). The ssGSEA approach also identified induction of IFN-γ response genes in intestinal stem cells/transit amplifying cells and enterocyte clusters 2 and 7 (Fig 2C).

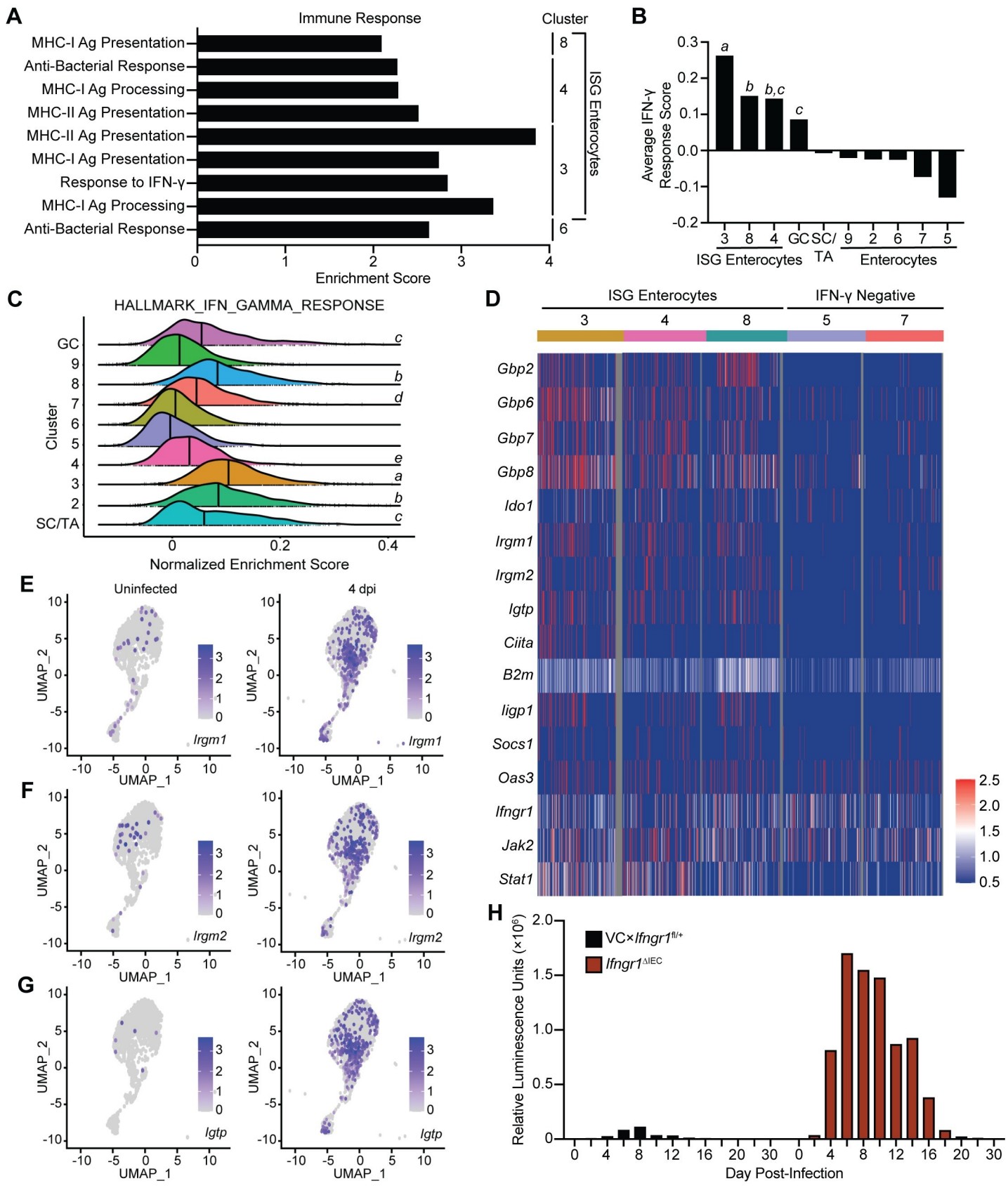

**Fig 2. IFN-γ signalling to IEC during *Cryptosporidium* infection limits parasite burden.** (**A**) Database for Annotation, Visualization and Integrated Discovery (DAVID) was used to generate functional annotation clustering of the marker genes for each cluster, functional annotations for the "Immune Response" category are indicated. (**B**) Average IFN-γ signature score (generated using expression of *Gbp7*, *Ifi47*, *Igtp*, *Iigp1*, *Irgm1*, *Tgtp1*, and *Zbp1*) for each IEC cluster. Letters indicate IFN-γ response score is significantly ($p < 0.0001$) higher than for other clusters. Clusters that share a letter are not significantly different from one another. (**C**) Single-sample gene set enrichment analysis (ssGSEA) was conducted on IEC clusters for the Hallmark IFN-γ Response gene set. Histograms of normalized enrichment scores are displayed; bold line indicates median score for each IEC cluster. Letters indicate IFN-γ response score is significantly ($p < 0.0001$) higher than for other clusters. Clusters that share a letter are not significantly different from one another. (**D**) Heatmap of gene expression in select IEC clusters for a curated list of IFN-γ-stimulated genes involved in cell-intrinsic control of infection and IFN-γ. Projection of (**E**) *Irgm1*, (**F**) *Irgm2*, and (**G**) *Igtp* (IRGM3) expression on UMAP of IEC clusters from uninfected and 4 dpi samples. (**H**) *Ifngr1*$^{\Delta IEC}$ and VCx*Ifngr1*$^{fl/+}$ heterozygote control mice were infected with 5x10$^4$ maCp oocysts and nanoluciferase activity was used to track fecal oocyst shedding. Abbreviations: stem cells/transit amplifying cells (SC/TA); goblet cells (GC); IFN-γ stimulated gene (ISG). Data in (**H**) are representative of two independent experiments, n = 3–5 mice per group, per experiment. Data in (**B**) and (**C**) were analyzed by one-way ANOVA with Tukey's post-test for multiple comparisons.

Next, the expression of specific ISGs that are involved in IFN-γ signalling and cell-intrinsic restriction of intracellular pathogens infection was analyzed. Expression of these genes was compared between the ISG enterocyte clusters and control clusters with negative IFN-γ response scores (IFN-γ negative clusters) (Fig 2D). Several genes, including *B2m*, *Ifngr1*, *Jak2*, or *Stat1* were constitutively expressed in enterocytes but enhanced in the IFN-γ responsive clusters (Fig 2D). In addition, several prototypical IFN-γ-stimulated anti-microbial effector proteins that belonged to the GBP family (*Gbp2*, *Gbp6*, *Gbp7*, and *Gbp8*), or the IRG family (*Irgm1*, *Irgm2*, and *Igtp*) [29] were associated with the ISG clusters (Fig 2D). Projection of these genes onto the scRNA-seq UMAP illustrated that they are induced 4 dpi and their expression overlaps with the ISG enterocyte clusters (Figs 2E–2G and S3A–S3D). Together, these analyses emphasize that *Cryptosporidium* infection results in the emergence of an IEC population associated with IFN-γ signalling.

While STAT1 signalling in IEC contributes to the control of *Cryptosporidium* [18] this transcription factor is activated by multiple cytokines that affect IEC and the role of IFN-γ in the activation of cell-intrinsic mechanisms to limit parasite replication *in vivo* has not yet been defined. Thus, to determine the impact of IFN-γ signalling to IEC, *Vil1*-Cre mice were crossed with mice that expressed a floxed allele of the IFN-γ receptor 1 chain to generate *Ifngr1*$^{\Delta IEC}$ mice in which IEC were unresponsive to IFN-γ. Relative to heterozygote control mice, which rapidly controlled the infection, by 4 dpi *Ifngr1*$^{\Delta IEC}$ mice had heightened levels of oocyst shedding that were sustained for 14 days (Fig 2H). Similar to *Ifng*$^{-/-}$ mice, *Ifngr1*$^{\Delta IEC}$ mice controlled parasite replication after approximately 20 days (Fig 2H), although parasite clearance observed in these animals is in contrast to persistent infection described in mice deficient for IFN-γ [3,19]. While a previous study demonstrated that *Stat1* in macrophages and dendritic cells is dispensable for parasite control [18], these data could suggest a role for IFN-γ signalling to immune cells in the clearance of *Cryptosporidium*. Nevertheless, these data sets demonstrate that IFN-γ acts directly on enterocytes to promote parasite control.

## *Cryptosporidium*-infected cells remain responsive to IFN-γ

The scRNA-seq analysis described above examines the global impact of *Cryptosporidium* on IEC responses but does not distinguish between uninfected and infected cells. To determine which IEC were infected, the reads from this data set were aligned to the *C. parvum* genome and 390 infected cells (that contained parasite transcripts) were identified 4 dpi (Fig 3A). No infection was detected in the stem cells/transit amplifying populations but rather parasites were detected predominantly in the ISG enterocytes and goblet cells (Fig 3A and 3B). Analysis of the distribution of infected cells across pseudotime similarly showed that parasites were mainly found in mid-villus and villus tip enterocytes (S4A Fig). In a separate experiment, the use of fluorescence microscopy confirmed the localization of parasites mid-villus and at the

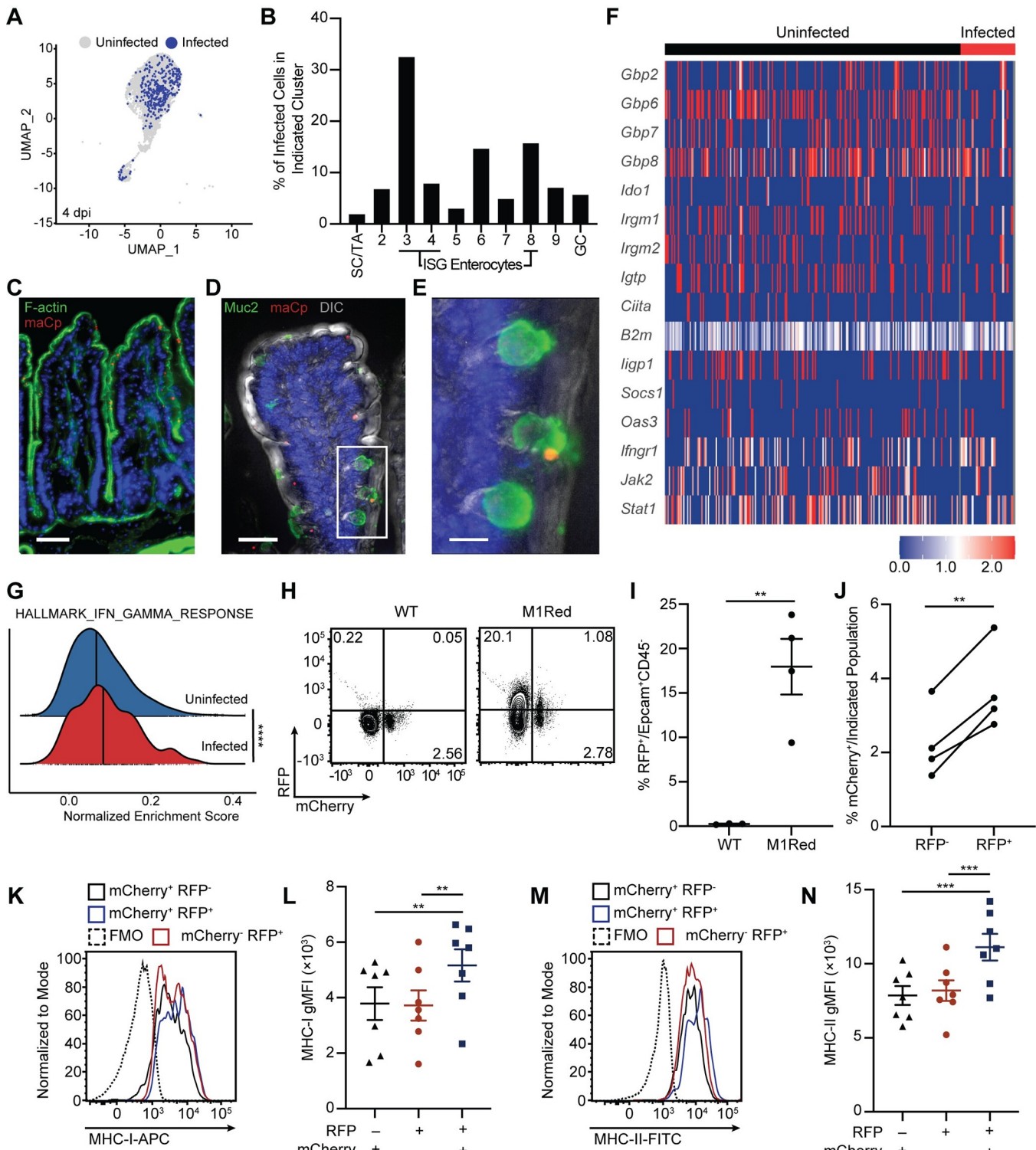

**Fig 3. IFN-γ-stimulated gene expression is upregulated in IEC during maCp infection.** (**A**) Projection of infected and uninfected cells on UMAP of IEC from the 4 dpi sample, based on the presence of *Cryptosporidium* transcripts. (**B**) Percentage of infected cells in enterocyte clusters. (**C**) Fluorescence microscopy image that shows the presence of IEC infected with *Cryptosporidium* (red) at the tip and along the sides of villi, with no parasites detected in the crypts. Staining also shows phalloidin stain for F-actin (green) and Hoechst nuclear stain (blue). Scale bar 50μm. (**D**) Fluorescence microscopy that shows a goblet cell (stained for Muc2 in green) infected with *Cryptosporidium* (red), with Hoechst nuclear stain (blue) and differential interference contrast (DIC; grey). White box is depicted in (**E**). Scale bar 30μm. (**E**) Inset of (**D**) showing close-up of infected goblet cell. Scale bar 10μm. (**F**) Heatmap of IFN-γ-stimulated gene

expression compared between uninfected and infected IEC from the 4 dpi sample. (**G**) Normalized enrichment score of the Hallmark IFN-γ Response gene set compared between uninfected and infected cells from the 4 dpi sample, as measured by ssGSEA. Bold line indicates median enrichment score for each group. (**H**) Representative flow plots and (**I**) frequency of RFP$^+$ IEC in WT and M1Red mice 5 dpi with maCp. (**J**) Frequency of infected RFP$^-$ and RFP$^+$ IEC from M1Red mice 5 dpi with maCp, as measured by mCherry. (**K**) Representative histogram and (**L**) gMFI of MHC-I expression on RFP$^-$mCherry$^+$, RFP$^+$mCherry$^-$, and RFP$^+$mCherry$^+$ IEC from M1Red mice 5 dpi with maCp. (**M-N**) Same as (**K-L**) for MHC-II. Data from (**H-J**) are representative of two independent experiments with n = 3–4 mice per group, per experiment. Data from (**K-N**) are pooled from two independent experiments with n = 3–4 mice per group, per experiment. Data in (**G**) and (**I**) were analyzed by two-tailed, unpaired Student's t-test. Data in (**J**) were analyzed by two-tailed, paired Student's t-test. Data in (**K**) and (**M**) were analyzed by one-way repeated measures ANOVA, with Geisser-Greenhouse correction and Tukey's post-test for multiple comparisons. Error bars represent mean ± SEM. \*\*$p < 0.01$, \*\*\*$p < 0.001$, \*\*\*\*$p < 0.0001$.

villus tip, as well as the presence of infected goblet cells (Fig 3C–3E). Interestingly, the percentage of infected cells within each IEC cluster correlated with the IFN-γ response score (S4B Fig) although uninfected and infected IEC showed comparable ISG upregulation (Fig 3F). Similarly, ssGSEA analysis demonstrated that infected IEC had a small, but significant, enrichment of the IFN-γ response gene set relative to uninfected cells (Fig 3G).

To assess if the transcriptional data sets described above correlate with translation of IFN-γ-target genes, M1Red mice were utilized, in which the *Irgm1* promoter drives *dsRed2* gene expression and provides a way to detect cells that are responsive to type I, II and III IFNs [30]. As such, M1Red mice infected with maCp provide the opportunity to concurrently monitor infection (mCherry$^+$) and response to IFN-γ (RFP$^+$). WT and M1Red mice were infected with maCp and reporter induction was analyzed 5 dpi, a time point when type I IFN is not detected and type III IFN has returned to baseline [20]. In these experiments, approximately 20% of IEC from infected M1Red mice were RFP$^+$ (Fig 3H and 3I) and a higher proportion of RFP$^+$ IEC were infected compared to RFP$^-$ IEC (Fig 3J). A comparison of surface expression of MHC-I and MHC-II, canonical IFN-γ response genes [31,32], highlighted that these proteins are upregulated during infection and their expression is highest on IEC from infected mice (S4C and S4D Fig). However, in infected M1Red mice, the expression of these proteins was highest on RFP$^+$mCherry$^+$ cells (Fig 3K–3N), which suggests that infected cells are most likely to be exposed to high levels of IFN-γ.

To directly test whether IEC infected with *Cryptosporidium* are responsive to IFN-γ, uninfected and infected *Ifng*$^{-/-}$ mice (5 dpi) were treated with PBS or IFN-γ and 12 hours later IEC were extracted for scRNA-seq. In initial studies, when *Ifng*$^{-/-}$ mice were treated with a single dose of 1μg or 5μg of murine IFN-γ 6h prior to infection, the 5μg dose resulted in parasite suppression for 24–48 hours (S5A Fig) and this dose was used in subsequent studies. It should be noted that a similar suppression was observed if IFN-γ was dosed multiple times either before or after infection (S5B and S5C Fig). In addition, treatment of *Stat1*$^{\Delta IEC}$ mice confirmed that restriction of infection was dependent on IFN-γ signalling to IEC (S5D Fig). Thus, this treatment regime allows control of the timing of IEC exposure to IFN-γ and permits direct comparison of gene expression between uninfected and infected cells in the same mice. A total of 42,521 cells were sequenced and after quality control there were 6,263 cells from the uninfected sample, 5,967 from the uninfected + IFN-γ sample, 6,666 from the infected sample and 5,479 from the infected + IFN-γ sample. Analysis of these data identified 14 IEC clusters (Fig 4A), including enterocytes, goblet cells, and additional IEC sub-types as above (S6A–S6F Fig). Infected cells were identified by the presence of parasite transcripts and analysis of their distribution largely overlapped between PBS- and IFN-γ-treated mice (Fig 4B). The proportion of infected cells found in each enterocyte cluster was significantly different following IFN-γ treatment, although the degree of infection correlated with the size of that cluster in each sample (Figs 4C and S6H).

Next, based on the earlier profile, heatmaps of select IFN-γ-stimulated genes were generated for each sample. For the uninfected group, IFN-γ treatment resulted in the induction of

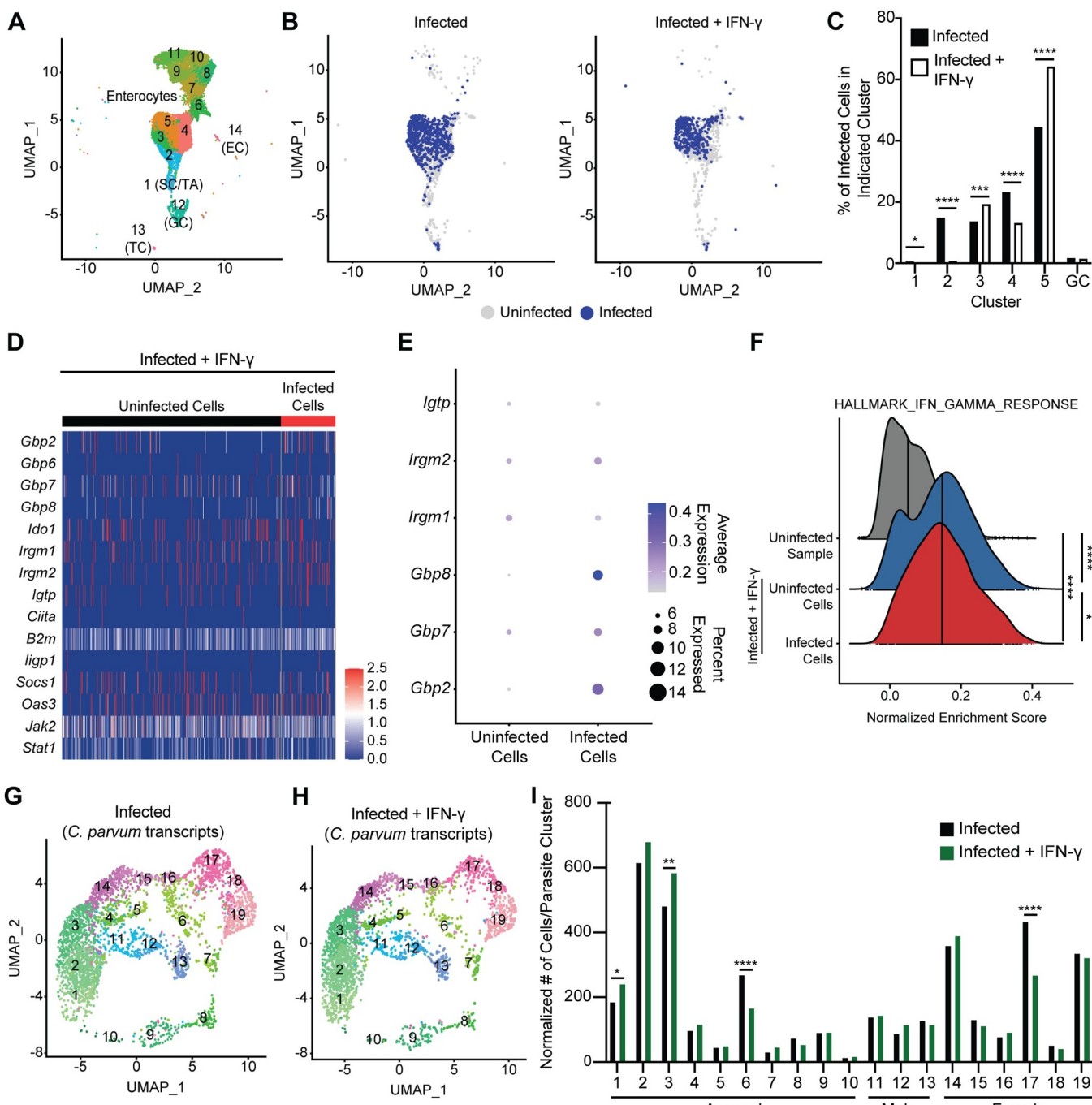

**Fig 4. Infected IEC remain responsive to IFN-γ stimulation.** (**A**) UMAP projection of scRNA-seq of IEC and IEL from uninfected and 5 dpi *Ifng*[-/-] mice treated with PBS or IFN-γ. (**B**) Projection of infected and uninfected cells on UMAP clustering from the infected *Ifng*[-/-] samples, based on the presence of *Cryptosporidium* transcripts. (**C**) Percentage of infected cells in enterocyte clusters for the infected *Ifng*[-/-] samples. (**D**) Heatmap of IFN-γ-stimulated gene expression compared between uninfected and infected IEC from the infected sample treated with IFN-γ. (**E**) Expression of select IFN-γ-stimulated genes was compared by dot plot between uninfected and infected cells from the infected sample treated with IFN-γ. (**F**) Normalized enrichment score of the Hallmark IFN-γ Response gene set compared between enterocytes from the uninfected sample (grey) and uninfected (blue) and infected (red) enterocytes from the infected sample treated with IFN-γ, as measured by ssGSEA. Vertical line indicates median enrichment score for each group. (**G-H**) UMAP of clustering analysis of scRNA-seq data aligned to the *Cryptosporidium* genome from *Ifng*[-/-] mice infected with maCp and treated with PBS (**G**) or IFN-γ (**H**). (**I**) Normalized number of cells per parasite cluster from each sample. Data in (**C**) and (**I**) were analyzed by χ² test for each cluster. Data in (**F**) were analyzed by one-way ANOVA with Tukey's post-test for multiple comparisons. \**p* < 0.05, \*\**p* < 0.01, \*\*\**p* < 0.001, \*\*\*\**p* < 0.0001.

several *Gbp* and *Irg* genes (S7A and S7B Fig). In the infected *Ifng*$^{-/-}$ mice, the induction of *Irg* genes was detectable in a small proportion of cells (S7C Fig) but ISG induction was most prominent after IFN-γ treatment (Fig 4D). This observation suggests there may be an infection-derived signal that "primes" IEC for IFN-γ responsiveness. Regardless, IFN-γ-mediated induction of this gene set was comparable between uninfected and infected cells (Fig 4D and 4E) and ssGSEA analysis demonstrated enrichment of the IFN-γ response hallmark that was increased relative to the uninfected sample, and nearly identical between uninfected and infected cells (Fig 4F). These data sets indicate that infected cells remain responsive to IFN-γ.

To determine whether IFN-γ treatment impacts parasite growth or development, a recent scRNA-seq analysis of the *Cryptosporidium* life cycle [33] was utilized. Data were aligned to the *C. parvum* genome and after quality control more parasites were recovered from the PBS-treated sample than the IFN-γ-treated sample (3,632 versus 2,676, respectively). To analyze parasite life stage distribution, these data were integrated with the sequencing data sets from the *Cryptosporidium* single cell atlas [33] for cluster analysis (S8A–S8C Fig). The UMAP projections show that both infected and infected + IFN-γ samples contain parasites distributed across clusters representing asexual (1–10, green clusters), male (11–13, blue clusters), and female (14–19, pink clusters) parasites (Figs 4G and 4H and S8D and S8E). The normalized distribution of infected cells across parasite clusters was similar in either sample (Fig 4I), which indicates that IFN-γ treatment does not appear to preferentially impact asexual, male, or female parasites.

## IFN-γ treatment promotes delayed control of *Cryptosporidium* infection

Since treatment with IFN-γ elicited a transcriptional response in IEC within 12 hours, experiments were performed to study the dynamics of IFN-γ-mediated parasite control. To determine how quickly IFN-γ restricts *Cryptosporidium in situ*, the impact of a single dose of IFN-γ on parasite burden in tissue and feces was analyzed. *Ifng*$^{-/-}$ mice were infected for 5 days with *maCp*, treated with PBS or IFN-γ, and parasite burden was assessed in ileal punch biopsies or feces at 6h, 12h, or 24h post-treatment. In the control group, parasite burden increased from the time of injection (T0) through the remainder of the time course (Fig 5A and 5B). IFN-γ-treated mice showed similar parasite burden to controls at 6h and 12h but a significant decrease in parasite burden occurred between 12h and 24h (Fig 5A and 5B). Flow cytometric analysis of the percentage of infected IEC also demonstrated a significantly reduced frequency of mCherry$^+$ IEC 24h post-treatment (Fig 5C and 5D). As such, despite an *in vivo* half-life of 1.1 minutes [34], a single treatment with IFN-γ elicited a transient, but delayed, reduction in *Cryptosporidium* burden.

In other experimental settings, IFN-γ rapidly activates macrophages to limit pathogen replication and does not require extended pre-exposure to cytokine [2]. Thus, one interpretation of the above kinetics is that the IEC response to IFN-γ appeared delayed. Indeed, when naïve M1Red mice were treated with a single dose of 5µg of IFN-γ, the use of flow cytometry and fluorescence microscopy revealed that in Epcam$^+$ IEC there was modest induction of RFP at 12h post-treatment but by 24h approximately half of all IEC expressed RFP before declining at 48h post-treatment (S9A–S9E Fig). Thus, the IEC response to IFN-γ stimulation *in vivo* peaks 24h post-stimulation.

To directly analyze whether IEC are primed by IFN-γ to control *Cryptosporidium*, organoids derived from the ilea of WT mice were used to generate IEC in an air-liquid interface (ALI) [35,36], which supports the *Cryptosporidium* life cycle [37]. When these cultures were infected with *C. parvum*, qPCR revealed increasing amounts of parasite genomic DNA at 24h to 48h post-infection (which represents 2–4 cycles of replication, lysis and reinfection), with

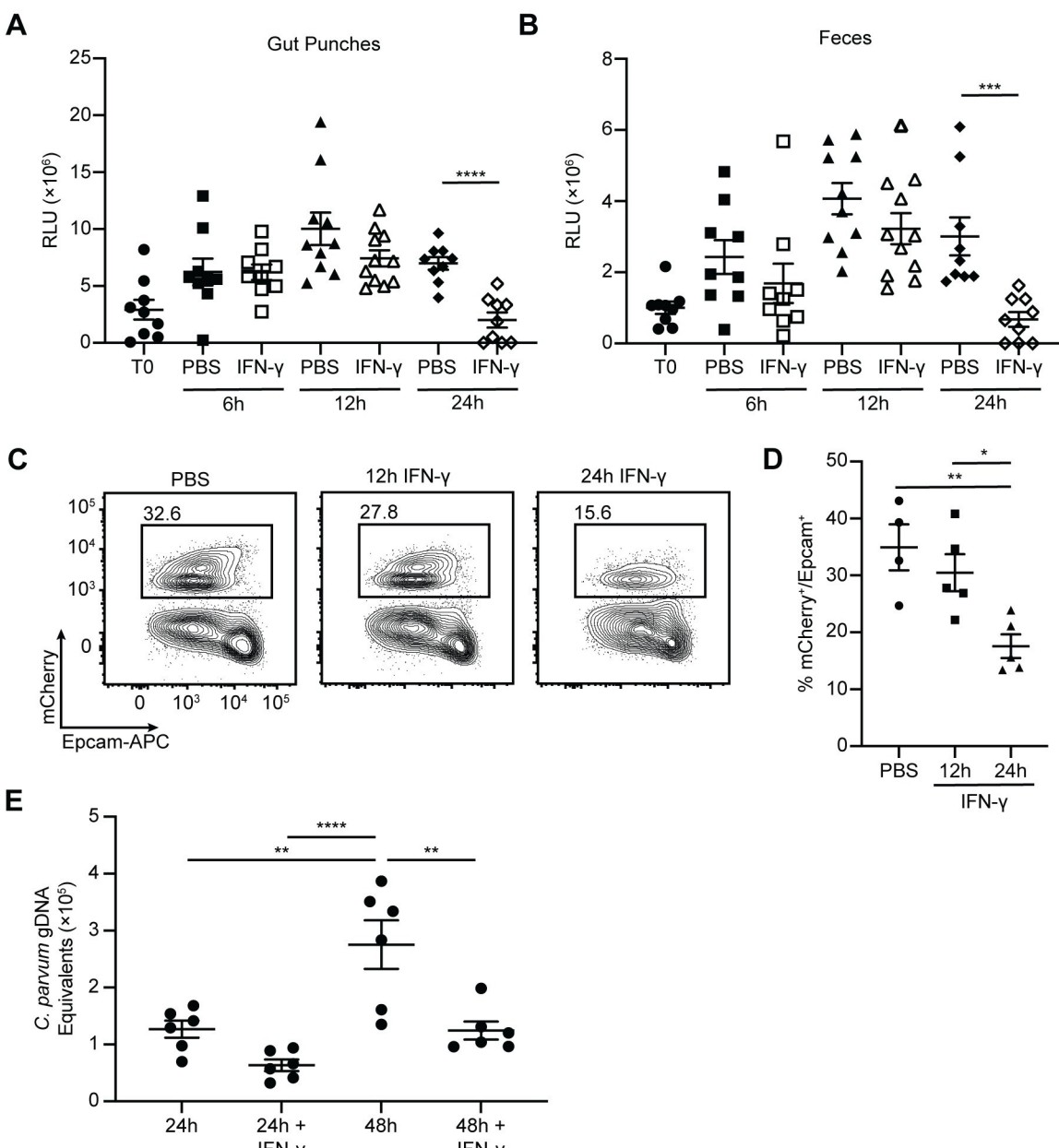

**Fig 5. IFN-γ treatment causes a delayed reduction in *Cryptosporidium* burden.** (**A**-**B**) *Ifng*[-/-] mice were infected with 10e[4] maCp oocysts and treated with PBS or IFN-γ 5 dpi. Parasite burden in punch biopsies (**A**) and oocyst shedding in feces (**B**) were measured by nanoluciferase activity at the time of treatment (T0) and 6, 12, and 24 hours post-treatment. (**C**-**D**) *Ifng*[-/-] mice were infected with 10[4] maCp oocysts and treated with PBS or IFN-γ 12 or 24 hours prior to being taken down for analysis 5 dpi. Representative flow plots (**C**) and frequency of infected cells (**D**) as measured by mCherry reporter expression in Epcam[+] IEC. (**E**) *C. parvum* burden was assessed by qPCR 24h and 48h post-infection in ALI monolayers with or without 24h pre-incubation with IFN-γ. Data are presented as genomic DNA (gDNA) equivalents after comparison to a standard curve of Ct value versus number of oocysts. Data in (**C**-**D**) are representative of two independent experiments, n = 3–5 mice per group, per experiment. Data in (**A**-**B**) are pooled from 3 independent experiments, n = 3–5 mice per group, per experiment and data in (**E**) are pooled from 2 independent experiments, n = 3 replicates per group, per experiment. Data in (**A**-**B**) were analyzed by two-tailed, unpaired Student's t-test at each time point. Data in (**C**-**E**) were analyzed by one-way ANOVA with Tukey's post-test for multiple comparisons. Error bars indicate mean ± SEM. $^*p < 0.05$, $^{**}p < 0.01$, $^{***}p < 0.001$, $^{****}p < 0.0001$.

minimal background detected in cultures incubated with heat-inactivated parasites (S9F Fig). When ALI monolayers were cultured with IFN-γ for 24h, followed by removal of cytokine and infection with *C. parvum*, there was no change in parasite burden at 24h post-infection, but there was a significant decrease in *C. parvum* gDNA equivalents at 48h post-infection (Fig 5E). Thus, prolonged exposure to IFN-γ primes IEC to activate cell-intrinsic mechanisms of parasite control.

## Discussion

Numerous reports have described histological changes in the gut associated with *Cryptosporidium* in experimental and natural models of infection that highlight increased intestinal stem cell turnover, crypt deepening and epithelial cell hyperplasia along the villus [18,19,38]. The data presented here leverage scRNA-seq to gain a better understanding of how *Cryptosporidium* infection and IFN-γ signalling impact infected and uninfected IEC. Perhaps the most prominent alterations associated with infection were the changes in mid-villus enterocytes, consistent with the reported hyperplasia of these cells. Increases in enterocyte frequency have also been observed using scRNA-seq during other intracellular infections (rotavirus and *Salmonella*) [39,40], while helminth infection induced goblet cell and tuft cell expansion [39], which illustrates infection-specific alterations to IEC populations. Single cell analysis of enterocytes during *Cryptosporidium* infection identified broad induction of immune response genes, in particular those involved in antigen processing and presentation and response to IFN-γ. Previous studies have also shown that MHC-II expression is increased on IEC exposed to IFN-γ [41,42], with pro- and anti-inflammatory roles ascribed to this pathway [41,43–46]. Still, IEC expression of MHC-I and MHC-II is consistent with the idea that IFN-γ stimulation of IEC promotes cognate interactions between CD4[+] and CD8[+] T cells and epithelial cells that are required for parasite control [5]. Indeed, a recent report highlighted that during infection with SFB, IEC expression of MHC-II was required for CD4[+] T cells to promote IEC turnover [47]. Goblet cells, which were infected by *Cryptosporidium* and had upregulated ISG expression, do not have a well-described role in control of *Cryptosporidium*, but in other experimental systems have been described to "pass" luminal antigen to local dendritic cells in the small intestine [48]. As such, the ability to genetically modify *Cryptosporidium* to express model antigens [17,49] will likely facilitate similar studies on antigen acquisition and presentation by various IEC populations.

There is currently a limited appreciation of the mechanisms used by IFN-γ to promote control of *Cryptosporidium* [5]. The identification of IFN-γ-stimulated genes expressed in infected cells, many of which are associated with anti-microbial activity in other model systems [50,51], provides a candidate list of effectors for additional studies. From this list, only IRGM1 and IRGM3 have been linked to control of *Cryptosporidium* [18]. Many of these genes are also upregulated in other systems where scRNA-seq has been used to study IEC responses. For example, during rotavirus infection there is upregulation of type I IFN response genes (many of which are also regulated by IFN-γ) in IEC [40] and expression of the anti-bacterial genes *Reg3b* and *Reg3g* are increased in *Salmonella*-infected mice [39]. The latter study also identified elevated expression of *Nlrp6* during *Salmonella* infection [39], which is upregulated in the scRNA-seq data presented here (S2 Table) and is a key initiator of the production of IL-18 required for innate immunity to *Cryptosporidium* [52]. At present, it remains a challenge to readily distinguish conserved IEC responses to inflammation associated with the production of IFNs versus those that are specific to different classes of pathogen.

IFN-γ has a prominent role in resistance to many intracellular bacterial and parasitic infections and this cytokine can activate haemopoietic and non-haemopoietic cells to limit

pathogen replication [2]. This is typically a rapid process in which a short pre-exposure, or even simultaneous addition of IFN-γ and pathogen is sufficient to restrict intracellular growth. Thus, previous studies with the M1Red reporter mice highlighted that following treatment with IFN-γ, high reporter activity is induced in monocytes after 6 hours and returns to baseline by 24 hours [30]. In contrast, in the studies presented here the administration of IFN-γ prior to or during infection limits *Cryptosporidium* burden, but requires between 12 and 24h, which correlates with maximal expression of the M1Red RFP reporter in IEC. These *in vivo* kinetics of IEC responsiveness to IFN-γ and the 12h *Cryptosporidium* life cycle [10] imply that for an IEC to restrict parasite replication, it needs to encounter IFN-γ prior to infection. A recent study demonstrated that IL-18 release by infected IEC and IL-12 production by dendritic cells stimulates gut-resident ILC1 to produce IFN-γ [18]. These data suggest a model in which areas of *Cryptosporidium* replication would be associated with local production of IFN-γ, which would be available to activate adjacent uninfected IEC. Indeed, in this study the strength of the IFN-γ signature in each IEC cluster correlated with the proportion of infected cells in that cluster. The observation that a single IFN-γ treatment prior to infection with *Cryptosporidium* delayed the onset of oocyst shedding further indicates that IFN-γ can act on uninfected IEC to limit parasite spread within the host. These data do not distinguish whether IFN-γ promotes death and shedding of infected cells, leads to direct death of the parasite or prevents the parasite from completion of its replication cycle and causes it to remain within the host cell until the IEC is shed into the lumen. The restriction of *Cryptosporidium* replication in IFN-γ-stimulated ALI monolayers suggests this system will enable formal testing of these possibilities, as similar studies in transformed cell lines are limited by the failure to support the complete parasite life cycle [11].

It has been proposed that the rapid replication cycle of *Cryptosporidium* represents a response to the rapid turnover of IEC [10]. It is also possible that the ability to invade, replicate and lyse IEC within 12h also provides a window for infected cells to evade restriction downstream of IFN-γ. There is precedence for a delayed responsiveness to IFN-γ in other cell types; while macrophages can be rapidly activated with IFN-γ to limit the replication of *T. gondii*, neurons must be pre-incubated with IFN-γ for 24h for parasite control [53,54]. In this context, low basal expression of STAT1 causes neurons to be comparatively "slow" to respond to IFN-γ, with maximal expression of *Stat1* and other IFN-γ-inducible genes occurring 24-48h post-stimulation [53,55]. Since IFN-γ can also cause cell death, delayed responsiveness to IFN-γ observed with IEC may be a host protective mechanism that reflects the immune tolerant state of the small intestine and could indicate a higher threshold of activation for IEC compared to other immune cells.

A broad theme in host-pathogen interactions is that the ability of microbes to disrupt relevant host immune pathways is important for pathogen success. Consistent with this idea, multiple intracellular pathogens have effectors that target the anti-microbial activities of IFN-γ. For example, *T. gondii* (a relative of *Cryptosporidium*) and rotavirus have effectors that disrupt IFN-γ- and STAT1-dependent transcription and signalling, and the function of antimicrobial effectors [56–59]. Since IFN-γ is critical for resistance to *Cryptosporidium*, we originally considered that infected cells might be hypo-responsive to the effects of IFN-γ. However, the data presented here indicate that infected cells remain responsive to IFN-γ, although this does not preclude the possibility that *Cryptosporidium* disrupts other aspects of IFN-γ-dependent immunity. An additional consideration may be the potential impact of parasite species and host specificity on immune evasion strategies. Different strains of *T. gondii* show dramatic differences in their ability to antagonize IFN-γ-mediated GBP recruitment to the PV [60]. Recent studies on the population structure of *Cryptosporidium* revealed adaptation to humans as a host and it seems likely that this was dependent on the development of species-specific

immune evasion strategies [60–62]. As such, the ability to compare *Cryptosporidium* species and strains that differ in virulence may provide new insights into host-specific interference with IFN-γ signalling while the ability to conduct genetic crosses between strains [63] offers the opportunity to uncover parasite genes and proteins that underlie these differences.

## Methods

### Ethics statement

All protocols for animal care were approved by the Institutional Animal Care and Use Committee of the University of Pennsylvania (Protocol #806292).

### Mice

C57BL/6J mice (stock #000664), *Ifng*$^{-/-}$ (stock #002287), *Vil1*-Cre (stock #004586) and *Ifngr1*$^{fl/fl}$ mice (stock #025394) were purchased from Jackson Laboratory and maintained in-house. *Stat1*$^{fl/fl}$ mice were generated as previously described [64] and maintained in-house. M1Red mice [30] were provided by Dr. Gregory Taylor (Duke University) and were maintained in-house. In-house breeding was performed to obtain all Cre-lox combinations. Mice used in this study were males or females ranging from 6 to 11 weeks. No differences were observed in infection burden between male and female mice. All mice were age matched within individual experiments.

### Parasites and infection

Transgenic *C. parvum* expressing nanoluciferase and mCherry [18] is propagated by orally infecting *Ifng*$^{-/-}$ mice. Oocysts are purified from fecal collections of infected mice using sucrose flotation followed by a cesium chloride gradient, as previously described [19]. Mice were infected with $1 \times 10^4$–$5 \times 10^4$ oocysts by oral gavage, diluted in a final volume of 100μL of PBS. To measure parasite burden in intestinal tissue, 5 mm biopsy punches were taken from the ileum and suspended in 1mL lysis buffer (50 mM tris HCl (pH 7.6), 2 mM DTT, 2 mM EDTA, 10% glycerol, and 1% TritonX in ddH$_2$O).

To quantify fecal oocyst shedding, 20mg fecal material was suspended in 1mL lysis buffer. Samples were shaken with glass beads for 5 minutes, then combined in a 1:1 ratio with Nano-Glo Luciferase solution (Promega, Ref N1150). A Promega GloMax plate reader was used to measure luminescence in technical triplicate for each sample. For tracking parasite burden over time, data are pooled cage-to-cage comparisons representing the average fecal parasite burden in each cage. The numbers of mice per experimental group and experimental replicates are provided in the figure legends and no data sets were excluded. Mice were randomly selected for experimental use and no blinding strategies were employed.

For ALI monolayer experiments, *C. parvum* (IOWA II strain, purchased from Bunchgrass Farms in Dreary, ID) was excysted prior to infection. Oocysts were pelleted and resuspended in 25% bleach in PBS on ice for 5 minutes and washed three times with PBS. Oocysts were then resuspended in 0.2 mM taurocholate (Sigma) in PBS for 10 minutes at 15˚C, washed three times, and resuspended in organoid growth medium for infections. For some experiments, sporozoites were excysted and heat-inactivated at 95˚C for 5 minutes.

### IFN-γ treatment

Mice were administered 1μg or 5μg of recombinant murine IFN-γ (Peprotech, catalog #315–05) diluted in a final volume of 200μL of PBS and injected intraperitoneally. Control mice were administered 200μL of PBS.

## IEC isolation

Mice were euthanized and the ileum (distal third of the small intestine) was harvested. Connective tissue and Peyer's patchers were removed, tissue was opened laterally, rinsed vigorously in ice-cold PBS and collected on ice into Hank's Balanced Salt Solution (HBSS) with 5% heat-inactivated fetal bovine serum (FBS) and 10mM HEPES. Single-cell suspensions of the IEC/IEL layer were prepared by transferring tissue to HBSS with 5% FBS, 5mM EDTA and 1mM DTT and shaking at 37˚C for 25 minutes, followed by two 1-minute washes in HBSS with 2mM EDTA and 10mM HEPES. Cell pellets were then resuspended and passed sequentially through 70μm and 40μm filters. We note that the extraction protocol used preferentially isolates IEC/IEL on the villus, while crypt-derived cells represent a smaller fraction of the single cell suspension.

## Flow cytometry

IEC were isolated as above and cells were stained with Ghost Dye Violet 510 viability dye (Tonbo; catalog #13-0870-T500) and surface stained with the following antibodies in an appropriate combination of fluorochromes: CD45.2 (BioLegend, clone 104), Epcam (BioLegend, clone G8.8), MHC-I (BioLegend, clone AF6-88.5), and MHC-II (BioLegend, clone M5/114.15.2). For experiments with the fluorescent reporters RFP or mCherry, samples were not fixed prior to analysis. Otherwise, samples were fixed on ice for 10 minutes in 2% PFA, washed, and resuspend in PBS with 2% bovine serum albumin prior to analysis. Data were collected on a LSR Fortessa (BD Biosciences) or a FACSymphony A3 (BD Biosciences) and analyzed with FlowJo v10 software (BD Life Sciences).

## Fluorescent imaging

Following infection or treatment with IFN-γ, mice were sacrificed and tissue from the distal third of the small intestine was flushed with ice-cold PBS. Tissue was opened laterally, rinsed again in ice-cold PBS, "swiss-rolled" and fixed for 1 hour in 2% paraformaldehyde in PBS at room temperature. Swiss rolls were transferred to 30% sucrose in PBS overnight at room temperature, followed by embedding in optimal cutting temperature (OCT) compound and storage at -80˚C. Tissues were sectioned, permeabilized for 45 minutes in hydration buffer (1% BSA and 0.1% Triton-x in PBS) and blocked for 45 minutes in saturation buffer (10% BSA and 0.1% Triton-x in PBS). Antibodies and stains included AF647-conjugated CD45 (BioLegend, clone 30-F11), polyclonal rabbit anti-Muc2 [C3] C-term (GeneTex), AF488-conjugated goat anti-rabbit IgG (H+L) (Invitrogen) and AF647-conjugated Phalloidin (Invitrogen) for F actin. Slides were stained with unconjugated primary antibodies diluted in hydration buffer for 2 hours, washed three times for 5 minutes in hydration buffer and stained for fluorescent antibodies or Phalloidin diluted in hydration buffer for 1 hour and 15 minutes. Nuclei were stained with Hoechst diluted in hydration buffer for 30 minutes, slides were washed three times for 5 minutes with hydration buffer and mounted using fluorogel (Electron Microscopy Science) mounting medium. All incubations were performed in a dark, humid chamber at room temperature. Slides were imaged with a Leica DM6000 Widefield microscope, with 3–5 representative images taken per section, and analyzed using Fiji software or Imaris software (Oxford Instruments).

## Single cell RNA sequencing

For scRNA-seq experiments, two mice were used for each group, and cells from both mice were pooled prior to sequencing. IEC/IEL single cell suspensions were prepared as above and

dead cells were removed using the Dead Cell Removal Kit (Miltenyi Biotec) using LS columns (Miltenyi Biotec), per the manufacturer's instructions. GEM encapsulation, reverse transcription, cleanup, and cDNA library preparation were done per manufacturer's instructions using the Chromium Next GEM Single Cell 3' Reagent Kits v3.1 (10X Genomics), Chromium Controller (10X Genomics) and C1000 Touch Thermal Cycler with 96-Deep Well Reaction Module (Bio-Rad). Fragment sizes and concentrations were quantified using High Sensitivity D5000 and High Sensitivity D1000 ScreenTape and TapeStation 4200 system (Agilent). Final library concentration was confirmed with Qubit 1X dsDNA HS Assay Kit and Qubit 3 (Invitrogen). Libraries were sequenced on a NextSeq 500 or NextSeq 2000 (Illumina).

Cell Ranger v7.0.0 was used to process sequencing reads and build a reference genome for *C. parvum* Iowa II (VEuPathDB, release 46). Processed reads from each sample were aligned to this reference genome or to the *Mus musculus* genome (GRCm38) to generate filtered feature-barcode matrices.

### Analysis of mouse scRNA-seq data

Filtered feature matrices were imported into R to make a Seurat object for each sample [65]. Samples within each experiment were merged and filtered to remove empty cells ($< 100$ features), doublets ($> 10,000$ UMI) and cells with a high frequency of mitochondrial genes ($>40\%$). Data were normalized and scaled, followed by principal component and jackstraw analyses to determine the dimensionality of the dataset and UMAP dimensional reduction. Differential gene expression analysis was performed using the FindAllMarkers function of Seurat. To identify pathways enriched within marker genes, functional enrichment clustering on the top 100 marker genes for each IEC cluster was performed using the Database for Annotation, Visualization and Integrated Discovery [66,67]. Functional enrichment clusters with an enrichment score of greater than 2 were selected, which corresponds to $p < 0.01$. Enrichment scores are calculated as the geometric mean of $-\log(p)$ for each member of the functional enrichment cluster. Pseudotime trajectory analysis was performed using monocle3 [68–70]. Single-sample GSEA was performed using escape [71].

### Analysis of C. parvum scRNA-seq data

Filtered feature matrices were imported into R to make a Seurat object for each sample [65]. Samples within each experiment were merged and filtered to remove empty cells ($< 100$ features), doublets ($> 1,200$ features or nCount $> 4,000$) and cells with greater than 60% of ribosomal genes (cgd2_1372, cgd2_1373, cgd3_665, cgd3_666, and cgd3_667). Cells that met these criteria were identified as infected in the mouse genome-aligned data sets. Data were normalized and variable features identified. Next, data were integrated with previously published scRNA-seq datasets from HCT-8 cells and *Ifng*[-/-] mice infected with *C. parvum* and sequenced at various time points post-infection [33]. Data were scaled, followed by principal component and jackstraw analyses to determine the dimensionality of the dataset and UMAP dimensional reduction. Male and female gene signatures were generated using previously defined gene lists (S3 Table) [33].

### 3T3 fibroblast cell culture and irradiation

NIH/3T3 mouse embryonic fibroblast cells (ATCC) were maintained at 37°C and 5% $CO_2$ in Dulbecco's Modified Eagle's Medium (DMEM) (ThermoFisher Scientific) with 10% fetal bovine serum (FBS; ATCC) and 1% penicillin/streptomycin (HyClone). For irradiation, cells were trypsinized (Fisher Scientific), suspended in growth medium and exposed to 3,000 rads

using an X-Rad 320 ix cabinet irradiator. Cells were aliquoted in 95% FBS and 5% DMSO and stored in liquid nitrogen until use.

## Intestinal organoids and Air-Liquid Interface

Organoids were established from the ilea of C57BL/6J mice (Jackson Laboratory, stock #000664) by following the Protocol described in the Technical Bulletin: Intestinal Epithelial Organoid Culture with IntestiCult Organoid Growth Medium (mouse) (StemCell Technologies, Document #28223). Organoids were maintained at 37˚C and 5% $CO_2$ as 3D spheroids in Matrigel (Corning) and organoid growth medium (50% L-WRN conditioned medium in DMEM with 10 μM ROCK inhibitor (Tocris Bioscience)), as previously described [35]. To establish Air-Liquid Interface monolayers, transwells (polyester membrane, 0.4 μm pores; Corning Costar) were coated with 10% Matrigel (Corning) in PBS for 20 minutes at 37˚C, after which excess was removed and transwells were seeded with $8x10^4$ irradiated 3T3 cells overnight in organoid growth medium. Mouse spheroids were collected from Matrigel domes and dissociated in 2 mL TrypLE (Fisher Scientific) for 5 minutes in a 37˚C water bath. Dissociated organoid cells were counted and transwells were seeded with $5x10^4$ IEC, with organoid growth medium present in both top and bottom compartments. Media was replenished every 3 days and on day 7 the medium in the top compartment was removed to establish the air-liquid interface. ALI monolayers were used for stimulation or infection 3 days after top media removal.

## Stimulation and infection of ALI monolayers

Once ALI monolayers were established, wells were stimulated with cells 250ng/mL recombinant murine IFN-γ (Peprotech) in the bottom compartment for 24h or with media alone. After 24h of stimulation, cytokine and media were removed, bottom compartments were washed with PBS, and fresh growth medium was added. ALI monolayers were infected by incubating $2\times10^5$ excysted *C. parvum* oocysts in the top compartment for 3h at 37˚C/5% $CO_2$, after which the infection media was removed. Each condition was performed in triplicate.

## Genomic DNA extraction and measurement of parasite burden

DNA was collected from transwells using the QIAamp DNA Mini Kit (Qiagen) at 24h or 48h post-infection. Cells were lysed by scraping into 100 μL Buffer ATL and 20 μL Proteinase K per sample and incubated overnight at 56˚C prior to column purification. DNA was eluted in 100 μL Buffer AE and diluted 1:10 in $H_2O$. For qPCR, 2μL of this dilution was used as template in a reaction with SsoAdvanced Universal SYBR Green Supermix (BioRad) per the manufacturer's instructions. Primers targeting *C. parvum* GAPDH are as follows: forward primer, 5'-CGG ATG GCC ATA CCT GTG AG-3'; reverse primer, 5'-GAA GAT GCG CTG GGA ACA AC-3'. A standard curve for *C. parvum* genomic DNA was generated by extracting DNA from a known quantity of *C. parvum* oocysts as above and creating a dilution series for qPCR. Reactions were performed on QuantStudio 5 System qPCR machine. Genomic DNA equivalents were determined by generating a standard curve for average Ct versus oocyst number in Excel (Microsoft).

## Statistics

Data were analyzed using GraphPad Prism 9 software. Specific tests for determining statistical significance are indicated in the figure legends and *p*-values of $< 0.05$ were considered statistically significant.

## Supporting information

**S1 Fig. Characterization of UMAP of scRNA-seq of uninfected, 4 dpi and 10 dpi WT mice.**
(**A**) Same UMAP as in Fig 1A with remaining clusters labelled. (**B-H**) Projection of gene
expression or gene signatures on UMAP of IEC and IEL clusters to identify those that repre-
sent (**B**) *Epcam*$^+$ IEC, (**C**) *Ptprc*$^+$ (CD45$^+$) IEL, (**D**) goblet cells, (**E**) tuft cells, (**F**) Paneth cells,
(**G**) enteroendocrine cells and (**H**) intestinal stem cells. (**I**) Cartoon depicting where cells from
each enterocyte cluster are projected to lie in the crypt and villus, coloured by pseudotime
value. (**J-M**) Graphs depicting expression of marker genes for crypt-adjacent (**J**), early mid-vil-
lus (**K**), late mid-villus (**L**) and villus tip (**M**) enterocytes versus pseudotime. (**N**) Fraction of
cells in clusters 14–24 separated by sample. (**O**) Dot plots of *Cd4*, *Cd8a*, and *Cd8b1* expression
in IEL clusters 15, 16, and 19. Data in (**N**) were analyzed by $\chi^2$ test between uninfected and 4
dpi and between uninfected and 10 dpi. $^*p < 0.05$, $^{**}p < 0.01$, $^{****}p < 0.0001$.
(PDF)

**S2 Fig. Differential gene expression in IEC clusters.** (**A**) Heatmap of top 100 marker genes
for each IEC cluster. List available in S1 Table. (**B-E**) Database for Annotation, Visualization
and Integrated Discovery (DAVID) was used to generate functional annotation clustering of
the marker genes for each cluster. Functional annotations for the (**B**) "Cellular Processes", (**C**)
"Mitochondria", (**D**) "Cell Structure", and (**E**) "Nutrient/Ion Transport" categories are indi-
cated. (**F**) IFN-γ signature score plotted on UMAP for enterocyte clusters, separated by time
point. (**G**) Deng *et al*. IFN-γ signature score (generated using expression of *Bst2*, *Stat1*, *Igtp*,
*Irf8*, *Ifit1*, *Ifit3*, *Tbk1*, *Parp14*, *Gbp7*, *Irf1*, *Trim6*, *Dcst1*, *Cd40*, *Rab43*, *Mrc1*, *Cited1*, *Ifngr2*, and
*Ido1*) plotted on the UMAP for each sample.
(PDF)

**S3 Fig. Selected *Gbp* expression from scRNA-seq dataset.** Expression of (**A**) *Gbp2*, (**B**) *Gbp6*,
(**C**) *Gbp7* and (**D**) *Gbp8* projected onto UMAP of IEC clusters from the uninfected and 4 dpi
samples.
(PDF)

**S4 Fig. MHC-I and MHC-II are upregulated on IEC in infected mice.** (**A**) Histogram of the
distribution of infected cells from the 4 dpi sample across pseudotime. (**B**) Linear regression
analysis of the percentage of infected cells in each enterocyte and goblet cell clusters versus the
average IFN-γ signature score for cells in that cluster. (**C-D**) Representative histogram and
gMFI of MHC-I (**C**) and MHC-II (**D**) in uninfected WT mice (black line), and WT mice
infected with 5x10$^4$ maCp oocysts (blue line). Dotted line on histograms indicates FMO con-
trol. (**E**) Frequency of mCherry$^-$RFP$^-$, mCherry$^+$RFP$^-$, mCherry$^+$RFP$^+$ and mCherry$^-$RFP$^+$ IEC
from infected M1Red mice 5 dpi with 5x10e$^4$ maCp oocysts. Data in (**C-E**) are representative
of two independent experiments with n = 4–5 mice per group per experiment. Data in (**C-D**)
were analyzed by two-tailed, unpaired Student's t-test. Error bars indicate mean ± SEM.
$^*p < 0.05$, $^{***}p < 0.001$.
(PDF)

**S5 Fig. Exogenous IFN-γ restricts *Cryptosporidium* infection by acting on IEC.** (**A**) *Ifng*$^{-/-}$
mice were treated with 1μg or 5μg of IFN-γ 6 hours prior to being infected with 10$^4$ maCp
oocysts and nanoluciferase activity was used to measure fecal oocyst shedding. (**B-C**) *Ifng*$^{-/-}$
mice were infected with 10$^4$ maCp oocysts and nanoluciferase activity was used to measure
fecal oocyst shedding. Mice were treated with PBS or IFN-γ 0, 1 and 2 dpi (**B**) or 5, 6, and 7
dpi (**C**). (**D**) *Stat1*$^{\Delta IEC}$ mice and VCx*Stat1*$^{fl/+}$ heterozygote control mice were infected with
5x10$^4$ maCp oocysts. One group of each genotype was treated with PBS or IFN-γ 0, 1 and 2 dpi

and nanoluciferase activity was used to measure fecal oocyst shedding. Data in (**A**-**D**) are representative of two independent experiments, n = 3–5 mice per group, per experiment.
(PDF)

**S6 Fig. Characterization of IEC clusters from scRNA-seq of *Ifng*<sup>-/-</sup> mice.** Projection of gene expression or gene signatures on UMAP clustering to identify clusters representing (**A**) *Epcam*<sup>+</sup> IEC, (**B**) goblet cells, (**C**) tuft cells, (**D**) Paneth cells, (**E**) enteroendocrine cells and (**F**) intestinal stem cells. (**G**) UMAP of IEC clusters colored by sample identity (blue, uninfected; purple, uninfected + IFN-γ; orange, infected; green, infected + IFN-γ). (**H**) Fraction of cells in IEC clusters, separated by sample. Data in (**F**) were analyzed by $\chi^2$ test between uninfected and uninfected + IFN-γ, and between infected and infected + IFN-γ. $^*p < 0.05$, $^{**}p < 0.01$, $^{***}p < 0.001$, $^{****}p < 0.0001$.
(PDF)

**S7 Fig. IFN-γ-stimulated gene induction in uninfected and *Cryptosporidium*-infected IEC.** Heatmap of IFN-γ-stimulated gene expression in uninfected IEC from the (**A**) uninfected and (**B**) uninfected + IFN-γ samples. (**C**) Heatmap of IFN-γ-stimulated gene expression compared between uninfected and infected IEC from the infected sample.
(PDF)

**S8 Fig. Identification of asexual, male and female parasites using the *Cryptosporidium* single cell atlas.** (**A**) UMAP clusters of scRNA-seq data from *Ifng*<sup>-/-</sup> mice infected with *C. parvum* (from Walzer et al.) [33] or maCp-infected mice treated with PBS (**B**) or IFN-γ (**C**) following alignment to the *Cryptosporidium* genome. (**D**) Male and (**E**) Female gene signature expression scores projected on UMAP clustering of scRNA-seq data from infected and infected + IFN-γ samples following alignment to *Cryptosporidium* genome. Genes used for each gene signature are in S3 Table.
(PDF)

**S9 Fig. IEC exhibit delayed responsiveness to IFN-γ stimulation.** (**A**) Fluorescence microscopy images of RFP (red), CD45 (grey) and Hoechst DNA stain (blue) in ileal sections from M1Red mice treated with PBS or 12h and 24h post-IFN-γ treatment. Scale bar 100 μm. White box in merge is depicted under "inset". (**B**) Representative flow cytometry plots and (**C**) frequency of RFP<sup>+</sup> Epcam<sup>+</sup> IEC and CD45<sup>+</sup> IEL from M1Red reporter mice treated with PBS or IFN-γ 12h and 24h prior to analysis. (**D**) Representative flow cytometry plots and (**E**) frequency of RFP<sup>+</sup> IEC 24h and 48h post-injection with IFN-γ. (**F**) *C. parvum* burden was assessed by qPCR 24h and 48h post-infection in ALI monolayers or in ALI monolayers infected with heat-inactivated *C. parvum* sporozoites. Data are presented as genomic DNA (gDNA) equivalents after comparison to a standard curve of Ct value versus number of oocysts. Data in (**A**-**E**) are representative of two independent experiments, with n = 3 mice per group, per experiment. Data in (**F**) are representative of two independent experiments with n = 3 replicates per group, per experiment. Data in (**C**) were analyzed by two-way ANOVA with Šidák's post-test for multiple comparisons. Data in (**E**) were analyzed by two-tailed, unpaired Student's t-test. Error bars indicate mean ± SEM. $^*p < 0.05$, $^{**}p < 0.01$, $^{****}p < 0.0001$.
(PDF)

**S1 Table. Top 100 marker genes for all clusters of WT scRNA-seq dataset.**
(XLSX)

**S2 Table. Top 100 marker genes for IEC clusters and DAVID pathway analysis from WT scRNA-seq dataset.**
(XLSX)

**S3 Table. Genes used to generate male and female gene signature scores in *C. parvum* genome-aligned data.**
(XLSX)

**S4 Table. Source data file.**
(XLSX)

## Acknowledgments

The authors gratefully acknowledge Dr. Gregory Taylor (Duke University) for providing M1Red mice and Dr. Sebastian Stifter, Dr. Lina Daniel, and Dr. Carl Feng (University of Sydney) for technical advice regarding these mice. Thank you to Elise Krespan, Clara Malekshahi and Dr. Daniel Beiting from the Center for Host-Microbial Interactions (University of Pennsylvania, RRID: SCR_022310) for technical assistance with scRNA-seq experiments and to Dr. Marilena Gentile and Dr. Andrew Vaughan for assistance with sample preparation for imaging. This study was supported by the Penn Cytomics and Cell Sorting Shared Resource Laboratory (RRID: SCR_022376), Penn Vet Comparative Pathology Core (RRID: SCR_022438), Penn Vet Imaging Core (RRID: SCR_022436), and Cell and Animal Radiation Core (RRID: SCR_022377), at the University of Pennsylvania. Fig 1C and S1I Fig were made using Biorender software. Thank you to members of the Hunter and Striepen Laboratories for critical feedback on the manuscript. Drs. Striepen and Hunter are supported by the Commonwealth of Pennsylvania.

## Author Contributions

**Conceptualization:** Ryan D. Pardy, Katelyn A. Walzer, Bethan A. Wallbank, Jessica H. Byerly, Boris Striepen, Christopher A. Hunter.

**Formal analysis:** Ryan D. Pardy, Katelyn A. Walzer, Bethan A. Wallbank, Jessica H. Byerly, Boris Striepen, Christopher A. Hunter.

**Funding acquisition:** Boris Striepen, Christopher A. Hunter.

**Investigation:** Ryan D. Pardy, Katelyn A. Walzer, Bethan A. Wallbank, Jessica H. Byerly, Keenan M. O'Dea, Ian S. Cohn, Breanne E. Haskins, Justin L. Roncaioli, Eleanor J. Smith, Gracyn Y. Buenconsejo.

**Methodology:** Ryan D. Pardy, Katelyn A. Walzer, Bethan A. Wallbank, Jessica H. Byerly, Keenan M. O'Dea, Ian S. Cohn, Breanne E. Haskins, Justin L. Roncaioli, Eleanor J. Smith, Gracyn Y. Buenconsejo, Boris Striepen, Christopher A. Hunter.

**Project administration:** Boris Striepen, Christopher A. Hunter.

**Resources:** Boris Striepen, Christopher A. Hunter.

**Supervision:** Boris Striepen, Christopher A. Hunter.

**Validation:** Ryan D. Pardy, Katelyn A. Walzer, Bethan A. Wallbank, Jessica H. Byerly, Keenan M. O'Dea, Ian S. Cohn, Breanne E. Haskins, Justin L. Roncaioli, Eleanor J. Smith, Gracyn Y. Buenconsejo, Boris Striepen, Christopher A. Hunter.

**Visualization:** Ryan D. Pardy.

**Writing – original draft:** Ryan D. Pardy, Christopher A. Hunter.

**Writing – review & editing:** Ryan D. Pardy, Katelyn A. Walzer, Bethan A. Wallbank, Jessica H. Byerly, Keenan M. O'Dea, Ian S. Cohn, Breanne E. Haskins, Justin L. Roncaioli, Eleanor J. Smith, Gracyn Y. Buenconsejo, Boris Striepen, Christopher A. Hunter.

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
