## [Decision Letter · Decision Letter 0]

7 Jan 2024

Dear Dr Hunter,

Thank you for submitting your manuscript “Analysis of intestinal epithelial cell responses to Cryptosporidium highlights the temporal effects of IFN-γ on parasite restriction” for consideration by PLOS Pathogens. Your article has been reviewed by three peer reviewers. All reviewers felt that your work contains interesting new information and is potentially acceptable for publication in PLOS Pathogens. However, the reviewers did have comments that require your consideration in the revision of your manuscript. Please pay particular attention to all the major points raised by the reviewers.

We cannot make any decision about publication until we have seen the revised manuscript and your response to the reviewers' comments. Your revised manuscript is also likely to be sent to reviewers for further evaluation.

Sincerely,

Philipp Olias

Academic Editor

PLOS Pathogens

Dominique Soldati-Favre

Section Editor

PLOS Pathogens

Kasturi Haldar

Editor-in-Chief

PLOS Pathogens

orcid.org/0000-0001-5065-158X

Michael Malim

Editor-in-Chief

PLOS Pathogens

orcid.org/0000-0002-7699-2064

Reviewer's Responses to Questions

**Part I - Summary**

Reviewer #1: The manuscript by Pardy et al. investigated the role of the IFN-g signalling in intestinal epithelial cells during the infection of mice with Cryptosporidium parasites. The authors very nicely desribe the kinetics of IFN-g responsiveness in different IEC clusters during the Cryptosporidium infection.

Overall, the study is well-designed and generally the main conclusions of the paper are supported by the data presented. I hope my comments will be helpful to potentially improve the manuscript further.

Reviewer #2: The manuscript by Pardy et al using single cell transcriptomics and functional in vivo to decipher the role of IFNg signaling in mediating epithelial responses during Cryptosporidium infection. Overall, this is a well-done study that is well suited to Plos pathogens. However, I have a few concerns that should be addressed prior to publication

Reviewer #3: The study reported in the manuscript entitled “Analysis of intestinal epithelial cell responses to Cryptosporidium highlights the temporal effects of IFNg on parasite restriction” by Pardy et al., reports the effects of IFNg on intestinal epithelial cells in response to infection by Cryptosporidium.

The study by Pardy et al. illuminates the important role of IFNγ-mediated response in intestinal epithelial cells to Cryptosporidium infection. The study utilizes single-cell RNA sequencing, aligning transcript reads to both the mouse and parasite genomes. This approach allows the analysis of infected and bystander epithelial cells. These observations suggest that IFN-γ activation led to a robust decrease in oocyst shedding in enterocytes and goblet cells. Overall, this study provides insight into the global impact of infection with Cryptosporidium on IEC and suggests a model in which IFN-γ-mediated bystander activation of uninfected enterocytes is important to limit the Cryptosporidium growth cycle within its host. However, some refinement of the text (see examples below) and further analysis need to be addressed before the manuscript is ready for publication.

**Part II – Major Issues: Key Experiments Required for Acceptance**

Reviewer #1: Major issue:

It would be of interest if the authors could suggest /show how IFN-g-mediated bystander activation of uninfected entrocytes can come about.

In the Figures 3E and 3F, it was shown that infected and uninfected IECs showed comparable enrichment of the IFN-g response gene set. Could it be possible that infected cells transmit signals for epithelial defense program via gap junction signaling (Ca2+ wave spreading etc.) to the uninfected cells during the Cryptosporidium infection?

Reviewer #2: • Throughout the paper enrichment scores for IFNg signaling are used without statistics to claim differences between groups.

• The claim of bystander activation both in the abstract and text based on scRNAseq data and enrichment scores seems exaggerated. IF imaging of STAT1 in RFP parasite infected mice should help corroborate this claim or it should be textually toned down.

• The IF images are extremely hard to interpret given the poor resolution and lack of negative (uninfected and isotype) controls.

• The text explaining figure 5 seems overly convoluted and could be simplified.

Reviewer #3: Comments:

1. A comprehensive enterocyte zonation analysis was conducted by Moor et al., 2018, showing the distribution of enterocyte function along the villus-crypt axis. While enterocytes at villus bottoms express an anti-bacterial gene program, the mid-villus enterocytes express absorption-related genes, and the villus-tip enterocytes express an anti-inflammatory program (the Cd73 immune-modulatory). How do these previous findings align with the pseudotime analysis performed in your study (shift in tasks upon infection)? And does the pseudotime reflect the enterocyte zonation correctly after the infection?

2. Related to this question, does the Cryptosporidium infection correlate with the mid-villus enterocytes reported in that study? And why does Crypto not infect the top villus compartment?

3. The proportion analysis lacks statistics and should show each mouse on the bars of each cell type to assess reproducibility.

4. The conclusion of the single cell analysis in Figures 1 and 2 (lines 189-190) needs to be refined or further examined. To me, it makes sense that the enterocytes gain IFNg program and are not generated or expended from the stem cells, as the proportion analysis showed no effect on stem/progenitor cells.

5. A known mechanism to prevent parasite infection is the “Weep and Sweep” phenomenon. A more comprehensive examination of the gut status is needed. Is the tissue more proliferative? And results in higher enterocyte differentiation? Using the single cell data, it is possible to examine and then validate it via tissue stains.

6. Related to the previous comment, do goblet cells secret more mucins to prevent the attachment of Crypto?

7. Parasites elicit type II immunity. Do you identify ILCs type II or Th2 cells under the infection? and any role of IL-13, IL-4, or IL-5 in the clearance of the crypto?

8. Please discuss the difference of Crypto in shifting T cells’ response to Type I and not Type II immunity.

9. What type of immune cells are elevated in IEL compartment?

**Part III – Minor Issues: Editorial and Data Presentation Modifications**

Reviewer #1: Minor issues:

When adjusting the manuscript, the authors may want to keep the following aspects in mind:

On the page 7, the authors mention that the number of T cells recruited to the epithelium was increased at day 10 post-infection. Did they analyzed the composition and potential heterogeneity of those T lymphocytes. Are there more CD4+ or CD8+ T cells recruited to the epithelial cells? Do they exhibit cytotoxic phenotype?

Several times, authors show the expansion of ISG enterocytes (clusters 2, 4 and 8) during Cryptosporidium infection, but there is also increase in IFN-g response score in goblet cells (cluster 10). Can authors speculate how important goblet cells (that are also associated with upregulated ISGs) for the clearance of the infection are? What about tuft cells?

On the page 10, authors show that the mice with defective IFN-g signaling in IECs displayed heightened levels of oocyst shedding, which was sustained for two weeks post-infection. In contrast to animals completely lacking IFN-g, mice lacking IFN-g signaling in IECs were able to clear infection. Does it mean that IFN-g signaling in macrophages and maybe in T cells is more essential to combat the Cryptosporidium infection? Could the authors discuss this possible “IFN-g-signalling hierarchy”?

Reviewer #2: • The figure presentations could be improved to ensure readability. For example, Figure 1 the scRNA seq clusters are labeled with numbers and only partially annotated. The genes used for annotation and not clearly noted and many clusters remain unannotated. Why not subcluster the epithelial cells and annotate?

• F images are also hard to interpret as

• The methods section is lacking any information about how epithelia were isolated and which fraction of epithelia (cypt/villi or both) were included in the analysis

• citations from the Yilmaz, Jabri and other lab have uncovered a role for MHCII expression in IECs should be included

Reviewer #3: Minor Comments:

1. Line 77, enterocytes express antimicrobial peptide as well.

2. Line 147, the figure number is wrong. Also, presenting the pseudotime analysis of the three conditions (Control and infected, day 4 and 10) would really help to show the enterocyte differentiation state in each condition.

3. Line 158, as with the epithelial cells, the change in cell proportions is need to be statistically evaluated. Which T cells are elevated and what about ILCs? I wonder if most T cells are Th1 and not Th2 cells.

4. The heatmaps in the study are hard to assess, so it may be beneficial to explore alternative data visualization tools, such as dot plots, to present specific genes.

5. Fig. 2D and E-G, maybe dot plots will serve you better, and a violin plot to show the signature?

6. As for the additional single cell paper (Ref. 29), what were the infection percentages (Figure 4B vs Figure 3A)? It seems higher than the one reported here and spans more enterocyte clusters. Why is that? Do these enterocyte states after infection correlate with this study?

7. As mentioned in the main comments, do the authors notice a change in epithelial proliferation or deepening of crypts, as stated in the discussion? These are important epithelial features associated with crypto and other gut infections that are not discussed in the results section. How does the IFNg treatment affect these features and the delayed enterocyte IFNg responses?

PLOS authors have the option to publish the peer review history of their article (what does this mean?). If published, this will include your full peer review and any attached files.

Reviewer #1: No

Reviewer #2: No

Reviewer #3: No
---

## [Decision Letter · Decision Letter 1]

14 Apr 2024

Dear Dr. Hunter,

We are pleased to inform you that your manuscript 'Analysis of intestinal epithelial cell responses to *Cryptosporidium* highlights the temporal effects of IFN-γ on parasite restriction' has been provisionally accepted for publication in PLOS Pathogens.

Best regards,

Philipp Olias

Academic Editor

PLOS Pathogens

Dominique Soldati-Favre

Section Editor

PLOS Pathogens

Michael Malim

Editor-in-Chief

PLOS Pathogens

orcid.org/0000-0002-7699-2064

---

## [Editor Report · Acceptance letter]

19 Apr 2024

Dear Dr. Hunter,

We are delighted to inform you that your manuscript, "Analysis of intestinal epithelial cell responses to *Cryptosporidium* highlights the temporal effects of IFN-γ on parasite restriction," has been formally accepted for publication in PLOS Pathogens.

Best regards,

Michael Malim

Editor-in-Chief

PLOS Pathogens

orcid.org/0000-0002-7699-2064